# Characteristics and Outcomes of Bloodstream Infections in a Tertiary-Care Pediatric Hematology–Oncology Unit: A 10-Year Study

**DOI:** 10.3390/jcm11030880

**Published:** 2022-02-08

**Authors:** Davide Mattei, Valentina Baretta, Annarita Mazzariol, Laura Maccacaro, Rita Balter, Ada Zaccaron, Elisa Bonetti, Matteo Chinello, Virginia Vitale, Giulia Caddeo, Maria Pia Esposto, Vincenza Pezzella, Davide Gibellini, Gloria Tridello, Simone Cesaro

**Affiliations:** 1Paediatric Hematology Oncology, Azienda Ospedaliera Universitaria Integrata, 37126 Verona, Italy; davide.mattei@studenti.univr.it (D.M.); valentina.baretta@aovr.veneto.it (V.B.); rita.balter@univr.it (R.B.); ada.zaccaron@aovr.veneto.it (A.Z.); elisa.bonetti2@aovr.veneto.it (E.B.); matteo.chinello@aovr.veneto.it (M.C.); virginia.vitale@aovr.veneto.it (V.V.); giulia.caddeo@aovr.veneto.it (G.C.); mariapia.esposto@studenti.univr.it (M.P.E.); vincenza.pezzella@aovr.veneto.it (V.P.); gloria.tridello@aovr.veneto.it (G.T.); 2Microbiology and Virology Unit, Department of Pathology, Azienda Ospedaliera Universitaria Integrata di Verona, 37126 Verona, Italy; annarita.mazzariol@univr.it (A.M.); laura.maccacaro@aovr.veneto.it (L.M.); davide.gibellini@univr.it (D.G.)

**Keywords:** neutropenia, neutropenic fever, bloodstream infection, bacteremia, empirical antibiotic therapy, multidrug resistance organism, Gram-positive, Gram-negative

## Abstract

Bloodstream infections (BSIs) after chemotherapy or hematopoietic stem cell transplantation (HSCT) are a leading cause of morbidity and mortality. Data on 154 BSIs that occurred in 111 onco-hematological patients (57 hematological malignancies, 28 solid tumors, and 26 non-malignant hematological diseases) were retrospectively collected and analyzed. Monomicrobial Gram-positive (GP), Gram-negative (GN), and fungal BSIs accounted for 50% (77/154), 38.3% (59/144), and 3.2% (5/154) of all episodes. Polymicrobial infections were 7.8% (12/154), while mixed bacterial–fungal infections were 0.6% (1/154). The most frequent GN isolates were *Escherichia coli* (46.9%), followed by *Pseudomonas aeruginosa* (21.9%), *Klebsiella* species (18.8%), and *Enterobacter* species (6.3%). Overall, 18.8% (12/64) of GN organisms were multidrug-resistant (seven *Escherichia coli*, three *Klebsiella pneumoniae*, and two *Enterobacter cloacae*), whereas GP resistance to glycopeptides was observed in 1% (1/97). Initial empirical antibiotic therapy was deemed inappropriate in 12.3% of BSIs (19/154). The 30-day mortality was 7.1% (11/154), while the bacteremia-attributable mortality was 3.9% (6/154). In multivariate analysis, septic shock was significantly associated with 30-day mortality (*p* = 0.0001). Attentive analysis of epidemiology and continuous microbiological surveillance are essential for the appropriate treatment of bacterial infections in pediatric onco-hematological patients.

## 1. Introduction

During recent decades, the improvement of treatments and supportive care of immunocompromised patients with hematological oncological diseases or undergoing stem cell transplantation determined a decrease in mortality and an increase in survival. The severe impairment of innate and adaptive immunity due to underlying disease and chemotherapy explains the high risk of infectious complications in these patients, BSIs being a major cause of morbidity and mortality [1,2,3]. Depending on the study, the incidence of BSIs is up to 50%, with an overall mortality >6% [4]. In patients with suspected BSIs, the timely initiation of empiric antibiotic therapy (EAT) is mandatory to reduce mortality. Knowledge of the local epidemiology of BSIs is pivotal to guide the correct selection of antibiotic therapy. In the last few decades, BSI epidemiology in hematological–oncological patients has changed, with a significant decrease in Gram-positive (GP) organisms and an increase in Gram-negative (GN) bacteria; more recently, a progressive rise in multidrug-resistant (MDR) strains has been reported [5,6,7,8]. The selection of suitable EAT in patients with febrile neutropenia (FN) represents a major challenge for physicians. In this study, we retrospectively assessed the clinical and microbiological data of all BSI episodes that occurred in a ten-year period at a nine-bed-equipped tertiary-care pediatric hematology–oncology unit to evaluate the etiology and outcome of BSIs, the incidence of MDR bacteria, and the change in EAT over time.

## 2. Materials and Methods

### 2.1. Setting, Study Population, and Design

This study analyzed all BSIs diagnosed in patients 0–18 years of age in the period between 1 January 2010 and 31 December 2019 that required hospitalization for intravenous broad-spectrum antibiotic therapy. Demographic, clinical, and microbiological characteristics of BSI episodes were collected by three authors (DM, VB, and SC) in an Excel database through the review of clinical charts. The review process was performed every 6 months as part of an internal infection control audit of nosocomial infections and MDR bacterial infections.

For each BSI episode the following data were recorded: age, gender, underlying disease, remission status of malignant diseases (diagnosis, relapse, complete remission, and partial remission), phase of treatment, type of antibacterial, antifungal and antiviral prophylaxis (if performed), presence and type of central venous catheter (CVC), presence of urinary catheter, neutrophil count, days of neutropenia before BSI, septic shock, intensive care unit (ICU) admission, type and source of bacteremia, antibiotic susceptibility profile, type of EAT, type of targeted antibiotic treatment, 30-day mortality, and overall survival.

In the case of repeated BSIs in the same patient, defervescence for at least 7 days and negative blood cultures were used as the criteria to classify a specific episode as a new episode.

### 2.2. Management of BSIs

Blood cultures from a peripheral vein and central line were drawn at hospital admission for each BSI episode together with total white blood cell count (WBC), absolute neutrophil count (ANC), C-reactive protein (CRP), and, in unstable patients, procalcitonin (PCT) concentrations. Clinical evaluation included a medical examination and determination of vital signs, such as temperature, blood pressure, and respiratory as well as heart rate. Blood cultures were routinely repeated after 72 h (on the third day) from the start of EAT and then repeated weekly if positive. 

Routine EAT was a combination of a third–fourth-generation cephalosporin (ceftazidime or cefepime) or an anti-pseudomonal penicillin (piperacilllin/tazobactam) with an aminoglycoside (amikacina), while a glycopeptide (vancomycin, teicoplanin) was added if there was a risk factor of Gram-positive infection (GPI), including the presence of CVCs, high-dose chemotherapy with cytarabine and fludarabine, and hematopoietic stem cell transplantation (HSCT).

Since 2016, all patients have undergone active surveillance for MDR bacterial colonization (extended-spectrum beta-lactamase (ESBL) or carbapenemase producers) by rectal swab at every hospital admission, for chemotherapy or other reasons, and repeated weekly in the case of prolonged hospitalization. In colonized patients the EAT was chosen on the basis of the potential antibiotic sensitivity profile of the colonizing bacteria until the result of the blood culture was known. Informed consent for data collection was obtained from parents, and patient data were managed according to Italian regulations for personal data protection. The study was approved by the local ethics committee.

### 2.3. Definitions

A BSI was defined as a blood culture (or cultures) positive for bacterial growth in at least one blood culture bottle, obtained from a patient with a compatible clinical syndrome (i.e., a patient who had evidence of one or more of the following signs and symptoms: fever (≥38 °C), hypothermia (<36 °C), chills, hypotension, oliguria, and/or elevated lactate levels). A polymicrobial BSI was defined as the isolation of more than one bacterial species from the same blood culture. A mixed BSI was considered as when bacteria and fungi were detected in the same sample.

GN organisms were classified as MDR if they were non-susceptible to three or more classes of antibiotics: third–fourth-generation cephalosporins, piperacillin/tazobactam, aminoglycosides, carbapenems, and fluoroquinolones [9]. Antibiotic resistance of GP bacteria was assessed according to the susceptibility to methicillin of *Staphylococcus (S.) aureus* and to glycopeptides of all GP bacteria [10].

Severe neutropenia was defined as an absolute neutrophil count (ANC) less than 0.5 × 10^9^/L [11]. The diagnosis of septic shock was based on hypotension (below the fifth percentile for age and sex) or the need for fluid resuscitation and/or inotropic support.

The 30-day mortality rate was defined as any death occurring within 30 days of the beginning of a BSI. Bacteremia-attributable mortality was defined as death directly related to bacteremia. The overall mortality rate was defined as any death recorded during the follow-up period, regardless of cause.

### 2.4. Assessment of Antibiotic Therapy

EAT was considered inappropriate when the isolated bacterium was not susceptible to any of the antibiotics used empirically in the first 96 h. After that, antibiotic treatment could continue unchanged or modified. If modified, it could either be escalated, by adding a new antibiotic or combination of antibiotics, or de-escalated, by discontinuing an antibiotic or combination of antibiotics.

### 2.5. Microbiology

Blood cultures were inoculated in bottles and processed in BACT/ALERT VIRTUO (bioMérieux, Florence, Italy). Negative bottles were discarded after 5 days of incubation, while positive bottles were examined through microscopic and culture methods. Bacterial and fungal growth on agar media were identified using MALDI-TOF VITEK MS (bioMérieux, Florence, Italy).

Antimicrobial susceptibility tests were performed using the VITEK2 system (bioMérieux, Florence, Italy), and results were interpreted following the EUCAST clinical breakpoints [12].

Multidrug-resistant microorganism screening was performed on rectal swabs. The specimen was streaked on selective media ChromoID-ESBL (bioMèrieux) with an ertapenem disk (10 µg) and on MacConkey agar with a meropenem disk (10 µg) in order to detect ESBL-producing enterobacteria, carbapenem-producing enterobacteria (CPE), and carbapenem-resistant *Pseudomonas* (*P.*) *aeruginosa*. ChromoID VRE was used to detect vancomycin-resistant enterococci (VRE). Carbapenemase production was confirmed with a Carba NP (#) rapid test and ESBL production with an ESBL NDP rapid test (#). Glycopeptide resistance of enterococci was confirmed by an E-test (bioMérieux, Lyon, France).

### 2.6. Statistical Analysis

The main characteristics of patients were reported by descriptive statistics, median, minimum, and maximum values were used for continuous variables, whilst absolute and percentage frequency were used for categorical variables. Comparisons between categorical variables were performed by the chi-square or Fisher’s exact test, as appropriate. The 30-, 90-, and 180-day as well as overall mortality were estimated by using the Kaplan–Meier method, considering death due to any cause as an event and the time from infection to the latest follow-up as survival time; differences between groups were tested by the logrank test. Univariate and multivariate risk factor analysis for overall survival were performed with the Cox regression model. The factors that were assessed were age, gender, type of underlying disease, remission status of malignant diseases, phase of treatment, type of antibacterial, antifungal and antiviral prophylaxis (if present), presence and type of central venous catheter (CVC), presence of urinary catheter, neutrophil count, days of neutropenia before BSI, duration of total neutropenia, septic shock, and intensive care unit (ICU) admission. Variables with a *p*-value < 0.1 at univariate analysis were entered into the multivariate model and selected according to a stepwise selection.

A *p*-value < 0.05 was considered statistically significant. All *p*-values were two-sided. All the analyses were performed using the statistical software SAS v. 9.4 (SAS Institute Inc., Cary, NC, USA).

## 3. Results

### 3.1. Demographics and Epidemiology

During the study period 154 BSI episodes were identified in 111 patients: 65 males and 46 females of a median age of 8.5 years (range: 0.3–18 years). The majority of patients, 34 (30.6%), were affected by acute lymphoblastic leukemia (ALL); 23 (20.7%) patients were affected by acute myeloid leukemia (AML), myelodysplastic syndrome (MDS), and chronic myeloid leukemia (CML). Figure 1 shows the distribution of underlying diseases.

A total of 55.5% (85/154) of these episodes occurred during the first-line treatment, 31.6% (49/154) in patients who underwent HSCT, and 12.9% (20/154) in patients who underwent chemotherapy for relapse. Table 1 describes the clinical characteristics of the cohort. The majority of patients had no definite focus of infection. The median peak temperature at the onset of the BSIs was 38.5 °C, ranging from 38.0 to 40.3 °C. Severe neutropenia was observed in 64.8% of the episodes (83/128; not reported in 26). The median number of days of severe neutropenia preceding the BSIs was 4 days, while the median duration of neutropenia was 17 days. One hundred and forty-four patients (93.5%) had a CVC inserted. Seventeen patients experienced septic shock (11%). Antibiotic, antifungal, and antiviral prophylaxis were already in place in 31.2% (48/154), 64.3% (99/154), and 39.6% (61/154) of episodes, respectively; all patients were on anti-*Pneumocystis jirovecii* prophylaxis.

### 3.2. Blood Culture Results

A total of 154 BSIs in 111 patients were recorded during the study period. Monomicrobial GP BSIs accounted for 50% (77/154) of all episodes, monomicrobial GN BSIs for 38.3% (59/144), and fungal organisms for 3.2% (5/154) of the total positive blood culture isolates. Polymicrobial infections accounted for 7.8% (12/154); 5.2% (8/154) caused by two GP microorganisms, 0.6% (1/154) caused by two GN bacteria, and 1.9% (3/154) caused by a mixed infection of a GP and a GN. Mixed bacterial–fungal infections accounted for 0.6% (1/154). Table 2 summarizes the most frequent causative agents of bloodstream infections.

The most frequent GN isolates were *Escherichia (E.) coli* (46.9%), followed by *P. aeruginosa* (21.9%), *Klebsiella (K.)* species (18.8%), and *Enterobacter* species (6.3%). A total of 28.1% (18/64) GN isolates belonged to the non-fermentative rods group: *Pseudomonas* species (14/64, 21.9%); *Acinetobacter* species (1/64); *Achromobacter xilososidans* (1/64, 1.6%); *Capnocytophaga* spp. (1/64, 1.6%); and another GN not identified at the level of species (1/64, 1.6%).

Overall, 18.8% (12/64) of GN organisms were MDR. Among these, 58.3% (7/12) were *E. coli*, 25% (3/12) were *Klebsiella pneumoniae*, and 16.7% (2/12) were *Enterobacter cloacae*. Table 3 summarizes the results of the antibiotic susceptibility of the most frequent GN isolates.

The resistance rate of *E. coli* was 30% for third–fourth-generation cephalosporins, 25% for semisynthetic penicillins/beta-lactamase inhibitors, 26.7% for aminoglycosides, and 53.3% for fluoroquinolones. As for *K. pneumoniae*, the resistance rate was 20.7% for third–fourth-generation cephalosporins, 41.7% for semisynthetic penicillins/beta-lactamase inhibitors, 16.7% for aminoglycosides, and 33.3% for fluoroquinolones.

The antibiotic susceptibility profile was available for nine of the fourteen *P. aeruginosa* strains isolated in this study (five missing); only one of these nine strains was resistant to third- and fourth-generation cephalosporins (11.1%), whereas the remaining eight strains were susceptible. Importantly, all GN isolates were susceptible to carbapenems.

The most common GP isolates were coagulase-negative staphylococci (CoNS) (61/97, 62.9%), with *S. epidermidis*, *S. haemolyticus*, and *S. hominis* being the most frequent. Ten isolates were *S. aureus* (10.3%). The remaining isolates are listed in Table 2. Overall, 2.1% (2/97) of the GP isolates were resistant to antibiotic treatment: one *S. aureus* was methicillin-resistant (1/7, 14.3%; antibiotic susceptibility test not available for three isolates) and one *S. epidermidis* was resistant to glycopeptides. *Enterococcus (E.) faecalis* and *E. faecium* accounted for 3.1% of the BSIs (3/97). No vancomycin-resistant enterococci (VRE) were isolated.

Among fungi, the most common isolate was *Candida (C.) parapsilosis* (4/6, 66.7%), followed by *Fusarium* species (1/6, 16.7%) and *Geotrichum capitatum* (1/6, 16.7%) (Table 2).

### 3.3. Empirical Antibiotic Treatment

All episodes were treated empirically with antibiotics: 140 out of 154 (90.9%) received an empirical combination therapy, 9 (5.8%) received an empirical monotherapy, while the data were missing for 5 episodes (3.2%). The most frequent empirical antibiotic therapy combinations were 52.6% (81/154) ceftazidime ± amikacin ± vancomycin/teicoplanin, 18.2% (28/154) meropenem ± amikacin ± vancomycin/teicoplanin, 3.2% (5/154) piperacillin/tazobactam ± vancomycin/teicoplanin, and 16.9% (26/154) other combinations (Appendix A). Based on microbiological results and clinical responses, the antibiotic therapy regimen after the first 96 h since the onset of BSIs was modified as follows: 46.8% (72/154) unchanged; 35.1% (54/154) stopped or de-escalated; 15.6% (24/154) escalated; and four were missing (Appendix A).

The mean duration of therapies was 18 days (range: 1–18) for ceftazidime/cefepime, 7 days (range: 2–29) for piperacillin/tazobactam, 17 days (range: 2–30) for meropenem, 11 days (range: 1–30) for amikacin, 9 days (range: 2–30) for vancomycin, and 7 days (range: 1–30) for teicoplanin.

EAT performed within 96 h from the beginning of BSIs was considered inappropriate in 12.3% of the BSIs (19/154).

### 3.4. Patient Outcomes

Septic shock occurred in 17 of 154 episodes (11%). The most common isolate in BSI episodes with septic shock was *E. coli* (5/17, 29.4%), followed by *K. pneumoniae* (3/17; 17.6%) and *Enterobacter cloacae* (2/17; 11.8%). Interestingly, three episodes of septic shock were caused by MDR GN organisms, whereas the other 14 episodes were caused by non-MDR GN organisms (nine episodes) and non-vancomycin-resistant GP bacteria (five episodes). 

The overall 30-day mortality was 7.1% (11/154). Six out of eleven deaths (3.9%) were due to complications directly related to bacteremia, two due to progression of disease (1.3%), two due to fungemia (1.3%), and one due to cytomegalovirus infection and disease (0.6%). The bacteremia-attributable mortality was 3.8% (6/154). Septic shock at BSI onset was significantly associated with a higher 30-day mortality in univariate (*p* = 0.0001) and multivariate analysis (odds ratio 17.9, *p* < 0.0001).

Among the patients who died from bacterial infections, two (1.3%) had a polymicrobial BSI. Specifically, one of these was caused by an ESBL-producing *E. coli* and *E. faecium* coinfection. The other polymicrobial BSI was caused by a coinfection of *K. pneumoniae* and a *P. aeruginosa*. In all these episodes EAT was appropriate. Three other deceased patients had a monomicrobial GN BSI caused by *P. aeruginosa*, ESBL-producing *K. pneumoniae*, and *E. coli*. EAT was appropriate also in all these BSIs. One patient had a monomicrobial GP BSI caused by an *E. faecium* that was susceptible to glycopeptides.

Fungi that were susceptible to all tested antifungal drugs caused the two remaining deaths: *Fusarium* and *C. parapsilosis*.

After a median follow-up of 1.68 years (confidence interval (CI) 95%, 0.99–2.53), the estimated 2-year overall survival (OS) was 67.8% (CI 95%, 58.2–75.7) (Figure 2). The overall mortality rate was 27.9% (43/154): nine patients (20.9%) died of bacterial (six), fungal (two), or viral infection (one) by 30 days from BSI onset. Among the remaining 34 patients (79.1%) who died, 28 died from disease progression and 6 HSCT patients died from transplant toxicity and multiorgan failure. 

## 4. Discussion

In onco-hematological patients, BSIs are a major cause of morbidity and mortality [13]. Neutropenia is the most important predisposing factor, and the prompt start of EAT, within a few hours from the onset of fever, is crucial for patient survival.

In the 1960s–1970s the trend in the global epidemiology of BSIs was characterized by the prevalence of GN bacteria. Since the mid-eighties GP strains have become predominant [14]. In 2000, with the diffuse use of fluoroquinolone prophylaxis, 76% of all BSIs in the United States was associated with GP microbes, of which CoNS, viridans streptococci, and enterococci were the most frequently isolated strains [6]. Lately, due to improvements in the overall handling of central venous catheters, the emergence of GN bacteria resistant to fluoroquinolones, and the diffusion of chemotherapy protocols associated with higher intestinal toxicity and endogenous bacteremia, a new increase in GN isolations has been documented [7,15,16]. In this study we found a higher proportion of GP isolates (60.2%) as compared to GN ones (39.8%). Among GP BSIs, CoNS were the most predominant organisms, followed by *S. aureus.* Among GN BSIs, *E. coli* was the most frequent isolate, followed by *P. aeruginosa*. These findings are consistent with other studies and may be explained, in part, by a lower pressure of GN selection on the intestinal flora in the absence of fluoroquinolone prophylaxis and by the large use of CVCs in the pediatric population [1,14,17,18].

Currently, the dramatic increase in infections caused by MDR organisms (MDROs) is a worldwide concern that makes effectively treating neutropenic fever even more challenging [1,5,8,15,17,19]. Major differences in antibiotic susceptibility may occur even within the same geographical area or within different areas of the same hospital [5]. Scrupulous knowledge of each particular epidemiological situation is crucial to use empirical antibiotics judiciously. In this study, 14 of 161 isolates (8.7%) presented antibiotic resistance of concern; of them, 12 were GN bacteria and two were GP bacteria. Considering the GN isolates, a moderately high rate of resistance to cephalosporins and antipseudomonal penicillins was found, being, respectively, 25% for ceftazidime/cefepime and 17.2% for piperacillin–tazobactam. While for cephalosporins the proportion of resistant strains is consistent with other studies, a slightly lower rate of piperacillin–tazobactam resistance was found (17.2% vs. 21.8%) [20]. The antibiotic resistance is worrisome when antibiotic monotherapy is used for FN, because it may result in an increased risk of treatment failure [21,22]. Due to the dramatic emergence of bacterial species carrying ESBL genes of antibiotic resistance, which inactivates third–fourth-generation cephalosporins and antipseudomonal penicillins, since the year 2000 the clinical use of carbapenems has increased [23,24]. This, in turn, caused an augmented number of bacteria isolates that produce carbapenemases with the ability to hydrolyze carbapenems [25,26,27]. We point out that, in contrast to other studies [28], all GNs of this study were susceptible to carbapenems. These variations between studies conducted at different institutions and countries highlight the importance of local surveillance, as antibiotic susceptibility patterns may not be broadly generalizable [28].

The rates of antibiotic resistance for *P. aeruginosa* are increasing worldwide. A large Italian study [29] of *P. aeruginosa* infections in pediatric oncology patients found a resistance rate of 33% to cefepime, 30% to ceftazidime, 27% to piperacillin, 25% to meropenem, 18% to ciprofloxacin, and of 11% to amikacin. Interestingly, we found resistance rates lower than other Italian centers: 11.1% resistance rates for cefepime/ceftazidime, 7.1% for piperacillin/tazobactam, and no resistance for amikacin and meropenem. Moreover, no *Pseudomonas* was MDR, compared to 31% in that study [29]. This underlines the significant epidemiological differences among centers of the same country or geographical area that, in turn, may reflect different compliances to hospital infection prevention measures or different clinical characteristics of the host.

Despite decades of well-performed clinical trials, no single empirical therapeutic regimen for the initial treatment of febrile patients with neutropenia has emerged as clearly superior to others [9,22,30,31]. Knowledge of local epidemiology can help choose the antibiotic components of EAT used to treat FN. This choice is different according to the policy used to manage FN. The escalation policy is defined as the use of a monotherapy that covers most GN bacteria, with the addition of one or more antibiotics in the case of clinical deterioration. This policy has the drawback of a limited coverage of GN and GP bacteria (e.g., CoNS, streptococci) and a high risk of failure in the case of bacterial antibiotic resistance [32,33]. The de-escalation policy is defined as administration from the beginning of a very broad-spectrum empirical regimen. Once microbiology results are available, the therapy is narrowed and continued until full neutrophil recovery and/or significant clinical improvement. Although these approaches are well-established in adult patients with cancer, particularly in those treated for severe sepsis in intensive care units [34,35], there are very few data on the escalation strategy in pediatric patients [36].

In this study, the most used EAT was the combination of ceftazidime ± amikacin ± vancomycin/teicoplanin (52.6%) or meropenem ± amikacin ± vancomycin/teicoplanin (18.2%). In most cases (46.8%) EAT was not changed after 96 h and continued unchanged, while the therapy was de-escalated in 35.1%; in just 15.6% of episodes EAT was escalated by adding a new antibiotic or shifting to new combinations [9]. This last finding is in line with the fact that most episodes received a broad-spectrum EAT from the beginning of the febrile episode.

A recent systematic review and meta-analysis including a total of 191 studies and 73.595 patients evidenced an association between the prevalence of any MDRO and advancing years with an inappropriate EAT. In addition, in recent studies the prevalence of MDROs, mainly multidrug-resistant GN bacteria, was significantly associated with the probability of prescribing an inappropriate EAT and higher mortality rates [37]. The low rate of GN MDROs and the preferred use of broad-spectrum antibiotic combinations in our cohort are probably the reasons why EAT resulted as being appropriate for GN bacteria in most episodes of FN. Regarding GP bacteria, the higher rate of inappropriate EAT compared to GN isolates was related to the fact that EAT usually comprised antibiotics directed against GN.

The 30-day mortality was 7.1%, with an attributable mortality caused by bacteremia of 3.9% (6/154). We believe that the low mortality observed in this study is a good result obtained through continuous microbiological surveillance, close clinical monitoring, rapid diagnosis, and prompt treatment in any suspected case of sepsis. 

## 5. Conclusions

This study showed the causative organisms of BSIs and their various antibiotic profiles in a group of patients with onco-hematological diseases or undergoing HSCT over a ten-year period. We found a limited incidence of MDROs during the study period as well as a low bacterial attributable mortality. We think that the measures adopted to contain bacterial infectious morbidity and mortality in our patient population, such as the adoption of hygienic measures (room isolation, hand washing, and personnel protective equipment), the monitoring of patients’ colonization, and prompt empirical antibiotic treatment tailored to the colonization status and the knowledge of the prevalent local epidemiology played a key role in obtaining these important results.

## Figures and Tables

**Figure 1 jcm-11-00880-f001:**
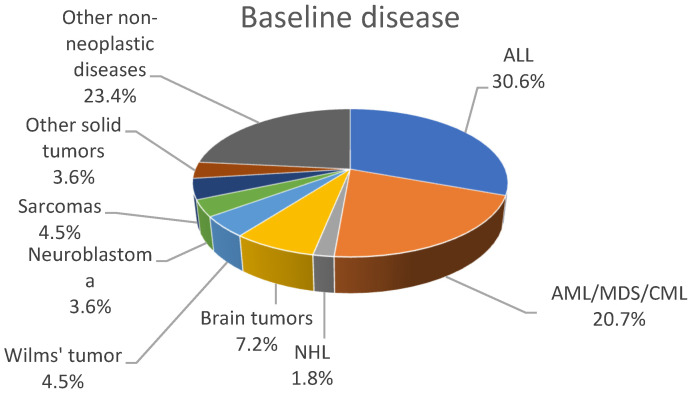
Distribution of underlying patient diseases.

**Figure 2 jcm-11-00880-f002:**
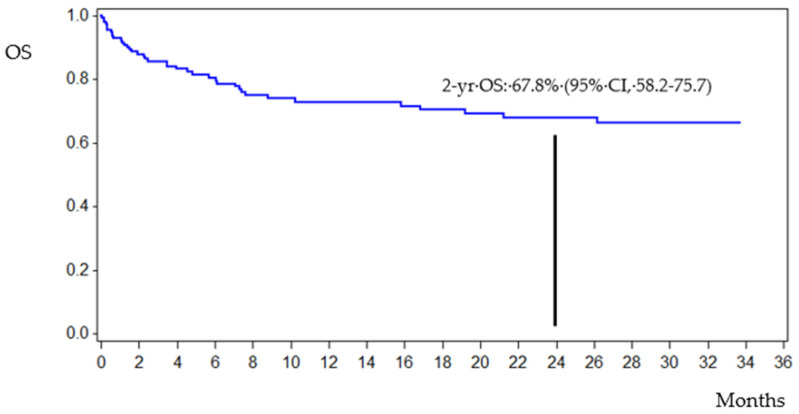
Survival curve.

**Table 1 jcm-11-00880-t001:** Clinical characteristics of 154 episodes of bloodstream infections (BSIs).

Clinical Characteristics	Episodes (*N* = 154), No. (%)
Peak temperature at the beginning of BSI	
Median (range)	38.5 °C (38.0–40.3)
Mean (SD)	38.74 (0.69)
N. obs	144
No. of neutrophils at BSI onset, cells/µL	
Median (range)	180 (0.0–43120.0)
<500	83 (64.8)
N. obs	128
Days of neutropenia before BSIs	
Median (range)	4 (6–240)
N. obs	92
Duration of total neutropenia	
Median (range)	17 (2.0–343.0)
Septic shock	17 (11.0)
Presence of central venous catheter	144 (93.5)
Antibiotic prophylaxis	48 (31.2)
Antifungal prophylaxis	99 (64.3)
Antiviral prophylaxis	61 (39.6)

**Table 2 jcm-11-00880-t002:** The most frequent causative agents of BSIs.

Gram-Positive	Episodes (*N* = 97), No. (%)
*Multidrug-resistant*	2 (2.1)
*Staphylococcus aureus*	10 (10.3)
CoNS	61 (62.9)
*Staphylococcus epidermidis*	29
*Staphylococcus haemolyticus*	7
Other staphylococcus species	
*S. hominis*	14
*S. warneri*	6
*S. lentus*	1
*S. saprophyticus*	1
*S. simulans*	1
*S. xylosus*	1
*S. capitis*	1
*Streptococcus mitis*	8 (8.2)
*Streptococcus salivarius*	3 (3.1)
*Streptococcus oralis*	2 (2.1)
*Enterococcus faecalis*	1 (1.0)
*Enterococcus faecium*	2 (2.1)
Other spp.	10 (10.3)
*Brevibacterium* spp.	1
*Aerococcus viridans*	1
*Rothia*	1
*Corynebacterium* spp.	2
*Micrococcus luteus*	5
**Gram-Negative**	**Episodes (*N* = 64), No. (%)**
*Multidrug resistant*	12 (18.8)
*Escherichia coli*	30 (46.9)
*Klebsiella* spp.	12 (18.8)
*Pseudomonas aeruginosa*	14 (21.9)
*Enterobacter* spp.	4 (6.3)
Other spp.	4 (6.3)
*Acinetobacter* spp.	1
*Achromobacter xilososidans*	1
*Capnocytophafa* spp.	1
*G-unspecified*	1
**Fungi**	**Episodes (*N* = 6), No. (%)**
*Candida parapsilosis*	4 (66.7)
*Fusarium* spp.	1 (16.7)
*Geotrichum capitatum*	1 (16.7)

**Table 3 jcm-11-00880-t003:** In vitro antibiotic resistance in the most frequent Gram-negative (GN) isolates.

GN Organisms	In Vitro Antibiotic Resistance No. (%)				
	Third- and Fourth-Generation Cephalosporins (%)	Semisynthetic Penicillins/β-Lactamase Inhibitors (%)	Aminoglycosides (%)	Carbapenems (%)	Fluoroquinolones (%)
*E. coli* (*N* = 30)	9 (30)	6 (20.7; 1 missing)	8 (26.7)	0	16 (53.3)
*K. pneumoniae* (*N* = 12)	3 (25)	5 (41.7)	2 (16.7)	0	4 (33.3)
*P. aeruginosa* (*N* = 9; 5 missing)	1 (11.1)	1 (7.1)	0	0	0

## Data Availability

The data presented in this study are available upon reasonable request from the corresponding author. The data are not publicly available due to privacy restrictions.

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
