# Peer review of "Characteristics and Outcomes of Bloodstream Infections in a Tertiary-Care Pediatric Hematology–Oncology Unit: A 10-Year Study"

_jcm, 2022, doi:10.3390/jcm11030880_

Round 1
Reviewer 1 Report
I have attached a PDF file including my comments. The topic is important and the data outlined is interesting, there are several formatting issues.

Reviewer 2 Report
The incidence of BSI was screened over a very long period of 10 years. features of the unit, hygiene conditions may have changed.It is unclear how long the patients received inpatient treatment.
Were all patients over the 10-year period included?
very small number of patients compared to this period.
the bed capacity of the unit etc. should be explained.
Did all patients have a central catheter?
What are the unit's blood culture protocol and indications?
Was a blood culture taken in all cases requiring hospitalization?
Reviewer 3 Report
This interesting study evaluated the epidemiology, risk factors and clinical outcomes of BSI in a pediatric hemato-oncology unit.
The study has been well conducted and the paper is well-written. It needs some minor spell-check of the microorganism names on table 2.
No further comments.
Round 2
Reviewer 2 Report
Author corrections are appropriate.